# Stiffness-aware neural network for learning Hamiltonian systems

**Senwei Liang**
Purdue University
liang339@purdue.edu

**Zhongzhan Huang**
Sun Yat-sen University
huangzhzh23@
mail2.sysu.edu.cn

**Hong Zhang** *
Argonne National Laboratory
hongzhang@anl.gov

## Abstract

We propose stiffness-aware neural network (SANN), a new method for learning Hamiltonian dynamical systems from data. SANN identifies and splits the training data into stiff and nonstiff portions based on a stiffness-aware index, a simple, yet effective metric we introduce to quantify the stiffness of the dynamical system. This classification along with a resampling technique allows us to apply different time integration strategies such as step size adaptation to better capture the dynamical characteristics of the Hamiltonian vector fields. We evaluate SANN on complex physical systems including a three-body problem and billiard model. We show that SANN is more stable and can better preserve energy when compared with the state-of-the-art methods, leading to significant improvement in accuracy.

## 1 Introduction

Data-driven modeling of dynamical systems provides a computationally inexpensive approach for exploiting the scientific law of the physical processes and predicting future phenomena (Giannakis & Majda, 2014; Harlim et al., 2021; Gu et al., 2021; Mou et al., 2021). With the superior approximation and generalization capacity (Lu et al., 2020; Shen et al., 2021) of neural networks (NNs), important advances have been made in learning Hamiltonian systems (Greydanus et al., 2019; Finzi et al., 2020; Tong et al., 2021). Based on the physical property that the total energy of the system (also called Hamiltonian) must be conserved, many studies model the Hamiltonian system by learning this conserved quantity from data (Greydanus et al., 2019; DiPietro et al., 2020). Even though some successes have been achieved, it remains challenging to learn the estimated system to capture the exact physical law, because of the elusive and chaotic characteristics of the systems (Choudhary et al., 2020; Marchal, 1990), especially for complex systems that are intrinsically stiff. When learning stiff dynamics, the NN optimization easily leads to an unstable solution or biased estimation due to the lack of constraint on parameters (Kim et al., 2021; Wang et al., 2020).

To illustrate the difficulty in learning stiff Hamiltonian dynamics, let us consider a three-body problem that describes the interactions of three particles under gravitational force. According to physical law, the repulsive force of two particles increases dramatically when they get close to each other. Hence, the close three-body interaction intensifies stiffness phenomena (Huang & Leimkuhler, 1997). Fig. 1 shows the reference orbits of three particles and the orbits learned by the Hamiltonian neural network (HNN) (Greydanus et al., 2019), a famous approach that directly approximates the Hamiltonian with a neural network. In Fig. 1(a, e), the HNN orbits coincide with the reference when the three particles are far from each other, which is consistent with the results in (Greydanus et al., 2019). However, the particles deviate from the reference orbits rapidly after close encounter, and the energy becomes nonconserved (Fig. 1(c, f)). This reveals although the NN is trained with the data containing close encounter, the NN does not fully capture the dynamics and easily diverges when stiffness grows. Two primary factors may affect the accuracy of the learned Hamiltonian.

**Implicit bias.** NN-based optimization has an implicit bias toward fitting a smooth function with the fast decay in the frequency domain (Xu et al., 2020; Cao et al., 2019). This implicit bias impedes the NN from capturing high-frequency components, such as singularities in an N-body problem.

---

*Correspondence should be addressed to hongzhang@anl.gov

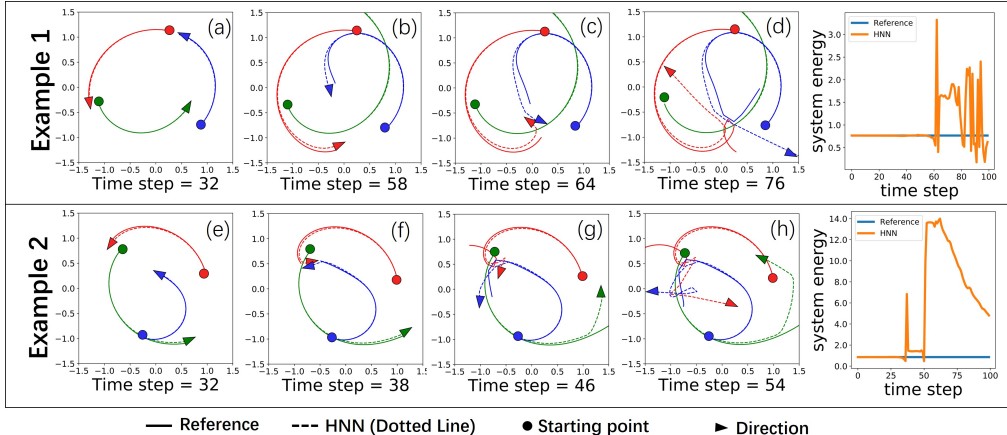

Figure 1: First four columns show a comparison of the reference orbits and the Hamiltonian neural network (HNN) orbits of three particles. The orbits of the different particles are displayed with different colors. The rightmost figures show the energy comparison. The reference orbits are simulated by the RKF45 solver with the ground truth Hamiltonian, and the energy is conserved. The energy of the HNN changes dramatically at the close interaction of two particles (see c, f), and eventually the HNN orbits diverge from the reference.

**Imbalanced stiffness proportion.** The stiffness of the dynamical system changes with time and varies across different trajectories (e.g., the same system with different initial conditions). It is not uncommon that only a small proportion of trajectories corresponds to stiff dynamics. As a statistical example, in the $1,000$ independent simulations of three-body trajectories following (Chen et al., 2019), 91.4% of the trajectories contain close encounter, but on average only $4.2\%$ of the time intervals within the trajectories contain close encounter.

To mitigate the implicit bias caused by NN-based optimization and the imbalanced stiffness proportion problem, we propose a new method called the stiffness-aware neural network (SANN) based on the stiffness classification of the training data. In our approach, we introduce a stiffness-aware index (SAI) as a simple, yet effective metric to classify the time intervals into stiff and nonstiff portions. For training efficiency, we integrate the Hamiltonian dynamics over different intervals with different step sizes based on their classification. To balance the ratio between the stiff group and the nonstiff group and avoid biased training, we resample the stiff intervals. Our contributions are as follows.

1. We identify the importance of the stiffness concept in learning Hamiltonian systems from the time series data. We show that SAI is easy to calculate and can be used to effectively determine stiffness intervals of the data.

2. We validate the SANN method with complex Hamiltonian dynamics including a three-body problem and billiard model. Extensive numerical results show that SANN can accurately predict the the stiff dynamics and significantly outperform the existing methods.

## 2 PRELIMINARIES

### 2.1 HAMILTONIAN SYSTEM

The Hamiltonian system describes the continuous-time evolution of states in the phase space $(\mathbf{p}, \mathbf{q})$, where $\mathbf{p} \in \mathbb{R}^n$ represents the generalized momentum and $\mathbf{q} \in \mathbb{R}^n$ denotes position coordinates. The Hamiltonian $\mathcal{H}(\mathbf{p}, \mathbf{q}) : \mathbb{R}^{2n} \to \mathbb{R}^1$ is denoted as the total energy of the system at $(\mathbf{p}, \mathbf{q})$. The dynamics can be described with $\mathcal{H}(\mathbf{p}, \mathbf{q})$ by

$$\frac{d\mathbf{q}}{dt} = \frac{\partial \mathcal{H}}{\partial \mathbf{p}}, \quad \frac{d\mathbf{p}}{dt} = -\frac{\partial \mathcal{H}}{\partial \mathbf{q}}. \tag{1}$$

With Eqn. (1), $\mathcal{H}(\mathbf{p}, \mathbf{q})$ is conservative during the evolution as $\frac{d\mathcal{H}}{dt} = \frac{\partial \mathcal{H}}{\partial \mathbf{p}} \cdot \frac{d\mathbf{p}}{dt} + \frac{\partial \mathcal{H}}{\partial \mathbf{q}} \cdot \frac{d\mathbf{q}}{dt} = 0$. In classical mechanics, the Hamiltonian is usually expressed as the sum of the kinetic and potential

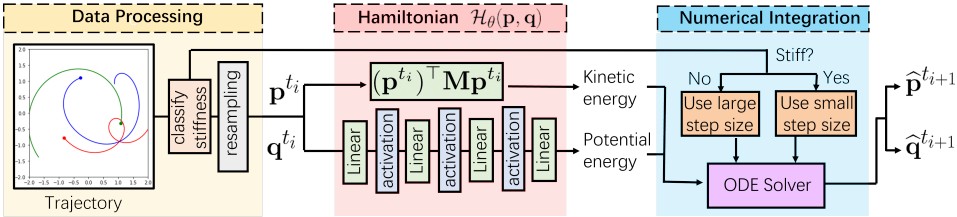

Figure 2: Workflow of SANN.

energies. We are interested in learning the separable Hamiltonian, namely, $\mathcal{H}(\mathbf{p}, \mathbf{q}) = T(\mathbf{p}) + V(\mathbf{q})$, where $T(\mathbf{p})$ is kinetic energy while $V(\mathbf{q})$ is potential energy. In this paper our goal is to learn the Hamiltonian of a dynamical system. What is available is the time series of observations $\{(\mathbf{p}^{t_i}, \mathbf{q}^{t_i})\}_{i=1}^N$. The Hamiltonian is parameterized by a neural network as follows,

$$\mathcal{H}(\mathbf{p}, \mathbf{q}) \approx \mathcal{H}_{\boldsymbol{\theta}}(\mathbf{p}, \mathbf{q}) \triangleq \mathbf{p}^\top \boldsymbol{M}\mathbf{p} + \phi(\mathbf{q}; \boldsymbol{W}), \tag{2}$$

where $\phi : \mathbb{R}^n \to \mathbb{R}^1$ is a fully connected NN, $\boldsymbol{M} \in \mathbb{R}^{n \times n}$ is a trainable matrix, and $\boldsymbol{\theta} \triangleq \{\boldsymbol{M}, \boldsymbol{W}\}$ is a set of all trainable parameters. Once we obtain $\mathcal{H}_{\boldsymbol{\theta}}$, the trajectories of the learned Hamiltonian system can be simulated by Eqn. (1) with an ordinary differential equation (ODE) solver.

## 2.2 ODE Solver for the Hamiltonian System

We use $\text{Forward}((\mathbf{p}^0, \mathbf{q}^0), \mathcal{H}, \Delta t)$ to denote a one-step integration of Eqn. (1) from $(\mathbf{p}^0, \mathbf{q}^0)$ over the step size $\Delta t$. The Euler method is a first-order Runge–Kutta method adopted in (Greydanus et al., 2019). When the Euler method is used, the integration becomes

$$(\mathbf{p}^{t_{i+1}}, \mathbf{q}^{t_{i+1}}) = \text{Forward}((\mathbf{p}^{t_i}, \mathbf{q}^{t_i}), \mathcal{H}, t_{i+1} - t_i) \triangleq (\mathbf{p}^{t_i}, \mathbf{q}^{t_i}) + (-\frac{\partial \mathcal{H}}{\partial \mathbf{q}^{t_i}}, \frac{\partial \mathcal{H}}{\partial \mathbf{p}^{t_i}})(t_{i+1} - t_i). \tag{3}$$

The Euler method is usually not suitable for stiff dynamics because of its poor stability property. Another popular numerical method (Fehlberg, 1969) for the Hamiltonian system is Runge—Kutta––Fehlberg (RKF45). RKF45 allows the step size to be changed adaptively based on an estimate of the local truncation error. This error-based step size control has great potential for improving computational efficiency. However, its effectiveness may be hampered by the use of minibatches, which are crucial for training. Minibatching adds an additional dimension to Eqn. (1). Controlling the step size requires considering a combined ODE system and estimating the error on all batch elements, as noted in Chen et al. (2018). Moreover, the step size is limited by the stiffest element in a batch, making it difficult to use a large step size especially when the batch size is large.

A symplectic integrator, a numerical method that conserves the energy quantity, is widely used in learning Hamiltonian system (Chen et al., 2019; Zhong et al., 2020). Leapfrog is a second-order symplectic integrator designed for a separable Hamiltonian. The Leapfrog scheme is in Appendix C.

## 3 Stiffness-Aware Neural Network

In this section we introduce our method, called SANN (stiffness-aware neural network), with its workflow shown in Fig. 2. First, we propose SAI (stiffness-aware index) to classify the time interval as either stiff or nonstiff. During training, we integrate the NN-parameterized Hamiltonian dynamics over the interval with different step sizes based on their classification. Next, we resample the stiff intervals to balance their ratio and avoid biased training.

### 3.1 Identifying the stiff interval

In this part we first discuss the stiffness index (SI) that is used to characterize stiffness for an ODE and propose the stiffness-aware index (SAI) to characterize the stiffness for the time series data.

**SI reveals the fastest rate of change of state.** We consider the dynamics of the state $\boldsymbol{u}$,

$$\frac{\mathrm{d}\boldsymbol{u}}{dt} = f(\boldsymbol{u}). \tag{4}$$

SI at the state $\boldsymbol{u}(t)$ is defined by $\max\{|Re(\lambda_i)|\}$, where $\lambda_i$ is the eigenvalue of the Jacobian matrix of Eqn. (4) (Aiken, 1985). To illustrate that SI reveals the fastest rate of change of state, we consider that Eqn. (4) is a linear system with constant coefficients, namely, $\frac{d\boldsymbol{u}}{dt} = \boldsymbol{A}\boldsymbol{u}$, and $\boldsymbol{A} \in \mathbb{R}^{n \times n}$ is a diagonalizable matrix with eigenvalues $\{\lambda_i\}_{i=1}^n$ and corresponding eigenvectors $\{\boldsymbol{v}_i\}_{i=1}^n$. Then the solution of the linear system is $\boldsymbol{u}(t) = \sum_{i=1}^n c_i \boldsymbol{v}_i e^{\lambda_i t}$. Let us suppose that $Re(\lambda_i) < 0, i = 1, ..., n$. We have $e^{\lambda_i t} \to \boldsymbol{0}$ as $t \to \infty$. Hence, $\max\{|Re(\lambda_i)|\}$ reveals the fastest speed of decaying to $\boldsymbol{0}$. If $\max\{|Re(\lambda_i)|\}$ is large, the integrator needs a small step size to reduce the local truncation error.

**Definition of SAI.** Given the observations $\{\boldsymbol{u}^{t_i}\}_{t=1}^N$ from the system (4), we define SAI at $\boldsymbol{u}^{t_i}$ by

$$\frac{1}{\|\boldsymbol{u}^{t_i}\|_2} \Big\| \frac{\boldsymbol{u}^{t_{i+1}} - \boldsymbol{u}^{t_i}}{t_{i+1} - t_i} \Big\|_2. \tag{5}$$

Intuitively, SAI characterizes the relative changing speed of the state $\boldsymbol{u}$ from time $t_i$ to $t_{i+1}$. The norm of finite difference $\left\| \frac{\boldsymbol{u}^{t_{i+1}} - \boldsymbol{u}^{t_i}}{t_{i+1} - t_i} \right\|_2$ approximates $\left\| f(\boldsymbol{u}^{t_i}) \right\|_2$, which is used in time parameterization of the adaptive time step method (Huang & Leimkuhler, 1997). In our data-driven setting, we cannot compute SI as the analytic expression of $f$ is unavailable. But SAI can serve as a proxy of SI in time series data, which will be demonstrated in Section 5.

Next, we split the training data into stiff and nonstiff portions by classifying the time interval as either stiff or nonstiff based on SAI. For a time series observation $\{\boldsymbol{u}^{t_i}\}_{i=1}^N$, we first compute SAI for each time interval. Let $\mathrm{SAI}_i = \frac{1}{\|\boldsymbol{u}^{t_i}\|_2} \left\| \frac{\boldsymbol{u}^{t_{i+1}} - \boldsymbol{u}^{t_i}}{t_{i+1} - t_i} \right\|_2$. We then rank $\{\mathrm{SAI}_i\}_{i=1}^{N-1}$. The interval $(t_i, t_{i+1})$ with a larger $\mathrm{SAI}_i$ is supposed to be a stiffer part. We use $\gamma \in (0, 1)$ to denote the stiff ratio, a hyperparameter to determine the ratio of the stiff portion. The intervals belonging to the top-$\gamma$ of $\{\mathrm{SAI}_i\}_{i=1}^{N-1}$ are identified as stiff intervals, and the others are classified as nonstiff. In other words, the intervals in the $(1 - \gamma) \times 100$-th percentile of SAI are identified as nonstiff intervals.

## 3.2 TRAINING THE ESTIMATED HAMILTONIAN

**Loss function.** Let $\{(\mathbf{p}^{t_i}, \mathbf{q}^{t_i})\}_{i=1}^N$ be the given trajectory from Hamiltonian system. Given a pair of consecutive states, $(\mathbf{p}^{t_i}, \mathbf{q}^{t_i})$ and $(\mathbf{p}^{t_{i+1}}, \mathbf{q}^{t_{i+1}})$, we integrate Eqn. (1) from $t_i$ to obtain the estimated solution $(\hat{\mathbf{p}}^{t_{i+1}}, \hat{\mathbf{q}}^{t_{i+1}})$ at $t_{i+1}$. To achieve a more accurate solution, we conduct the integration over $[t_i, t_{i+1}]$ with a small time step $\frac{t_{i+1} - t_i}{S}$, where $S$ is an integer hyperparameter called partition. Specifically, we proceed as follows:

$$(\hat{\mathbf{p}}^{t_{i+s/S}}, \hat{\mathbf{q}}^{t_{i+s/S}}) = \mathrm{Forward}((\hat{\mathbf{p}}^{t_{i+(s-1)/S}}, \hat{\mathbf{q}}^{t_{i+(s-1)/S}}), \mathcal{H}_{\boldsymbol{\theta}}, \frac{t_{i+1} - t_i}{S}), s = 1, \cdots, S, \tag{6}$$

where $(\hat{\mathbf{p}}^{t_i}, \hat{\mathbf{q}}^{t_i}) \triangleq (\mathbf{p}^{t_i}, \mathbf{q}^{t_i})$. For the stiff equation, the step size of the numerical integration must be small enough for the stable solution. Therefore, the large partition $S$ should be chosen for the stiff portion while the small partition $S$ is used for the nonstiff portion. The parameter $\boldsymbol{\theta}$ in $\mathcal{H}_{\boldsymbol{\theta}}$ can be optimized by minimizing the mean squared error between the prediction and the ground truth:

$$\frac{1}{N-1} \sum_{i=1}^{N-1} \|\mathbf{p}^{t_{i+1}} - \hat{\mathbf{p}}^{t_{i+1}}\|_2^2 + \|\mathbf{q}^{t_{i+1}} - \hat{\mathbf{q}}^{t_{i+1}}\|_2^2, \tag{7}$$

with stochastic optimization algorithms such as Adam (Kingma & Ba, 2014).

**Resampling.** As demonstrated in Section 1, the stiff phenomena may comprise only a small proportion in the trajectory of the Hamiltonian system. We usually pick the stiff ratio $\gamma$ with a value less than 0.5. To avoid biased regression and unstable training caused by imbalanced categories of intervals, we adopt a random resampling technique that balances the data by replicating the stiff intervals $K$ times. A typical choice for replication $K$ is $\frac{1-\gamma}{\gamma}$ such that the number of the stiff intervals is comparable to that of the nonstiff intervals.

## 4 EXPERIMENTS

To evaluate the performance of SANN, we use two complex Hamiltonian systems: the billiard model and the three-body problem. We compare SANN with two famous approaches, including

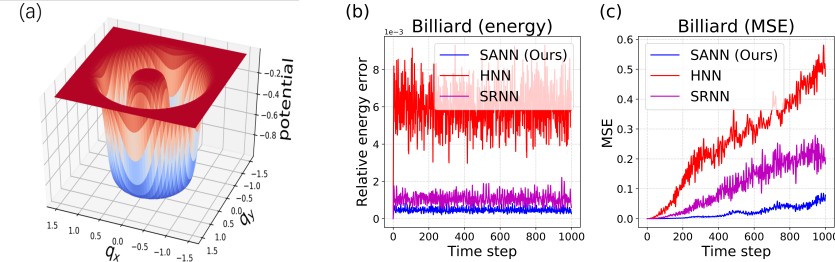

Figure 3: (a) Profile of the potential function for the billiard model; $\mathbf{q} = (q_x, q_y) \in \mathbb{R}^2$ denotes the position. Comparison of (b) relative energy error and (c) MSE for billiard model.

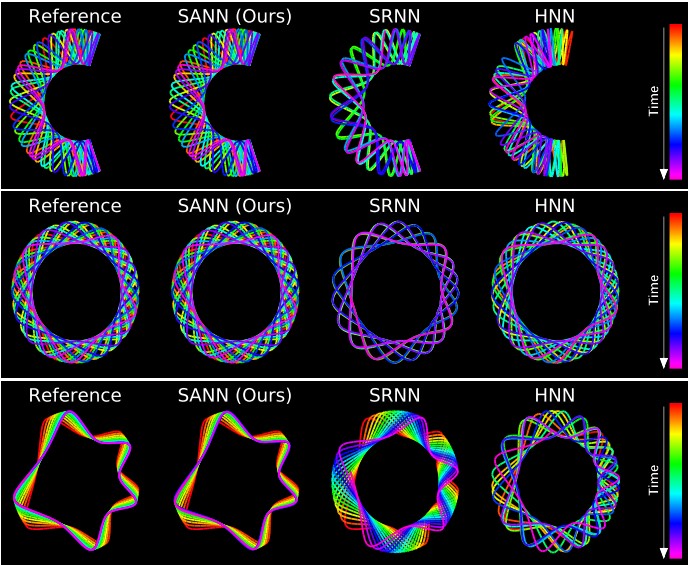

Figure 4: Comparison of billiard orbits simulated using the Hamiltonian learned by different methods. We can see that SANN produces orbits that are almost identical to those of the reference. We provide more results in Appendix E and dynamic graphs on website.

HNN (Greydanus et al., 2019) and SRNN (Chen et al., 2019). All approaches aim to learn $\mathcal{H}_{\boldsymbol{\theta}}(\mathbf{p}, \mathbf{q})$ in Eqn. (1) as a representation of the dynamics. HNN trains $\mathcal{H}_{\boldsymbol{\theta}}(\mathbf{p}, \mathbf{q})$ by minimizing the error between the partial derivatives of $\mathcal{H}_{\boldsymbol{\theta}}(\mathbf{p}, \mathbf{q})$ and time derivatives approximated by data. SRNN utilizes symplectic solvers to conduct multistep integration and trains $\mathcal{H}_{\boldsymbol{\theta}}(\mathbf{p}, \mathbf{q})$ by minimizing the error between the prediction and the data. SANN uses a similar loss function but with stiffness awareness as shown in Section 3.2. For a fair comparison, we use the Leapfrog solver for both SRNN and SANN. To evaluate the accuracy of the learned $\mathcal{H}_{\boldsymbol{\theta}}(\mathbf{p}, \mathbf{q})$ for different methods, we simulate the dynamics using the Leapfrog solver with the same settings during testing. We provide a guidance for hyperparameter selection in Appendix G and additional experiments on the Pendulum-N problem which involves inseparable Hamiltonian in Appendix H.

## 4.1 BILLIARD MODEL

The billiard model has wide application in real-world physical systems, spanning quantum-classical correspondence (Stöckmann & Stein, 1990), lasers (Stone, 2010), quantum dots (Ponomarenko et al., 2008), and nanodevices (Chen et al., 2016). The billiard model that we consider describes a billiard bouncing between a ring with soft boundaries (Choudhary et al., 2020). The Hamiltonian is defined as

$$\mathcal{H}(\mathbf{p}, \mathbf{q}) = \frac{1}{2}\|\mathbf{p}\|_2^2 + \left(1 + \exp\left(\frac{r_\alpha - \|\mathbf{q}\|_2}{s}\right)\right)^{-1} - \left(1 + \exp\left(\frac{r_\beta - \|\mathbf{q} - (q_0, 0)\|_2}{s}\right)\right)^{-1}, \quad (8)$$

where $\mathbf{p} \in \mathbb{R}^2$ and $\mathbf{q} \in \mathbb{R}^2$ are the momentum and the position of a billiard on a 2D plane, respectively; $r_\alpha, r_\beta$ are the radius of the outer circle and the inner circle, respectively; $s$ is the softness of the boundary; and $(q_0, 0)$ is a shift of the inner circle from the center. As shown in Fig. 3(a), the potential function becomes sharp near the boundaries of the ring, causing a rapid change of the momenta of the billiard ball when it gets close to the boundaries.

**Experimental setup.** The parameters $r_\alpha, r_\beta, s$, and $q_0$ are set to be $1.0, 0.5, 0.05$, and $0.1$, respectively. We use the physics-informed model $\mathcal{H}_{\boldsymbol{\theta}}(\mathbf{p}, \mathbf{q})$ defined in Eqn. (2); $\phi$ is a 3-hidden-layer fully connected NN with width 256 per layer and a rational activation function (Boullé et al., 2020). We simulate the training set with 100 trajectories and the testing set with 30 trajectories by RKF45. Each trajectory contains 1,000 time steps with step size of 0.1. We set the stiff ratio $\gamma$ to be 10% and the partition $S$ to be 10 for the stiff interval and 2 for the nonstiff interval. The loss function (7) is optimized by Adam with a batch size of 1,024, and we use an initial learning rate of 0.001 for 500 epochs. The learning rate follows cosine decay with the increasing training epoch. During testing, we integrate ODE (1) using the Leapfrog integrator with a fixed step size of 0.005.

**Results.** SANN can predict the dynamics more accurately than HNN and SRNN can. Fig. 3(b, c) shows the mean square error (MSE) with time and relative energy error, which is defined as the ratio of difference between the current energy and the initial energy to the initial energy. We can see that SANN maintains a much smaller energy error compared with HNN or SRNN. On average, SANN reduces the relative energy error by 56.07% compared with SRNN and 92.13% compared with HNN. The gap in terms of MSE enlarges as the number of time steps increases. At the last time step, the MSE of SANN is 56.91% lower than that of SRNN and 83.44% lower than that of HNN. Fig. 4 gives a comparison of the predicted trajectories. SANN recovers visually the same trajectories as the reference, while the trajectories obtained with SRNN and HNN clearly diverge from the reference. Additional results for noisy data are given in Appendix F. A discussion on implicit bias is given in Appendix A.

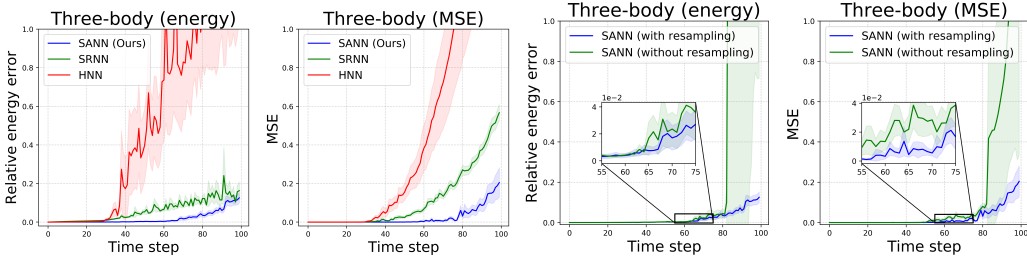

Figure 5: The first two figures show the comparison of relative energy error and MSE with varied time for the three-body problem. The last two figures show the comparison of performance with and without resampling.

## 4.2 THREE-BODY PROBLEM

We consider a three-body problem that describes the motion of three particles under Newtonian gravitational force. The Hamiltonian of the three-body system is given by

$$\mathcal{H}(\mathbf{p}, \mathbf{q}) = \frac{1}{2} \sum_{i=1}^{3} \frac{\|\mathbf{p}_i\|_2^2}{m_i} + \sum_{1 \le i < j \le 3} -\frac{Gm_i m_j}{\|\mathbf{q}_i - \mathbf{q}_j\|_2}, \tag{9}$$

where $\mathbf{p}_i \in \mathbb{R}^2$, $\mathbf{q}_i \in \mathbb{R}^2$ and $m_i$ are the momentum, position, and mass of the $i$th particle, $i = 1, 2, 3$, $\mathbf{p} \triangleq (\mathbf{p}_1, \mathbf{p}_2, \mathbf{p}_3)$ and $\mathbf{q} \triangleq (\mathbf{q}_1, \mathbf{q}_2, \mathbf{q}_3)$, and $G$ is the gravitational constant. One can see that the potential function incurs singularities when the distance between any two particles is small.

**Experiment setup.** We set $m_1 = m_2 = m_3 = 1$ and $G = 1$ in our experiments. To approximate the Hamiltonian of three-body dynamics, we use a physics-informed $\mathcal{H}_{\boldsymbol{\theta}}$ as follows,

$$\mathcal{H}_{\boldsymbol{\theta}}(\mathbf{p}, \mathbf{q}) \triangleq \mathbf{p}^\top \boldsymbol{M} \mathbf{p} + \phi(\|\mathbf{q}_1 - \mathbf{q}_2\|_2, \|\mathbf{q}_1 - \mathbf{q}_3\|_2, \|\mathbf{q}_2 - \mathbf{q}_3\|_2; \boldsymbol{W}), \tag{10}$$

where $\phi$ is defined as in Section 4.1, and we incorporate the physics information where the potential is a function of pairwise distances. The training data comprises 1,000 trajectories, and the length

of each trajectory $N$ is 60 with the step size 0.1. The random initialization of initial states follows (Chen et al., 2019). The testing data set consists of 300 trajectories with length $N$ of 100 and step size of 0.1. We set the stiff ratio $\gamma$ to be 10% and the partition $S$ to be 50 for the stiff interval and 10 for the nonstiff interval. When training, we minimize the loss function (7) by Adam with the batch size of 1,024. $\mathcal{H}_\theta$ is trained with an initial learning rate of 0.001; the learning rate decays for 3,000 epochs. During evaluation, the Leapfrog integrator with fixed step size of 0.01 is used.

**Results.** SANN learns a more accurate Hamiltonian than HNN and SRNN do on the three-body problem. Fig. 6 shows the orbits of three particles simulated by different methods and a system energy comparison. The relative energy error and MSE of orbits simulated by different methods are presented in Fig. 5. In Fig. 6, in the beginning the orbits of all methods coincide with the reference. However, we see that the orbits of HNN and SRNN rapidly drift away from the reference, and the energy changes dramatically whenever two orbits meet. On the contrary, SANN produces orbits that are nearly identical to the orbits of the reference, and the energy remains roughly constant. From Fig. 5, we see that SANN achieves lower energy error and MSE compared with HNN and SRNN. On average, SANN reduces the relative energy error by 67.80% compared with SRNN and 96.30% compared with HNN. Also, at the last time step SANN's MSE is 63.84% lower than SRNN's and 92.65% lower than HNN's.

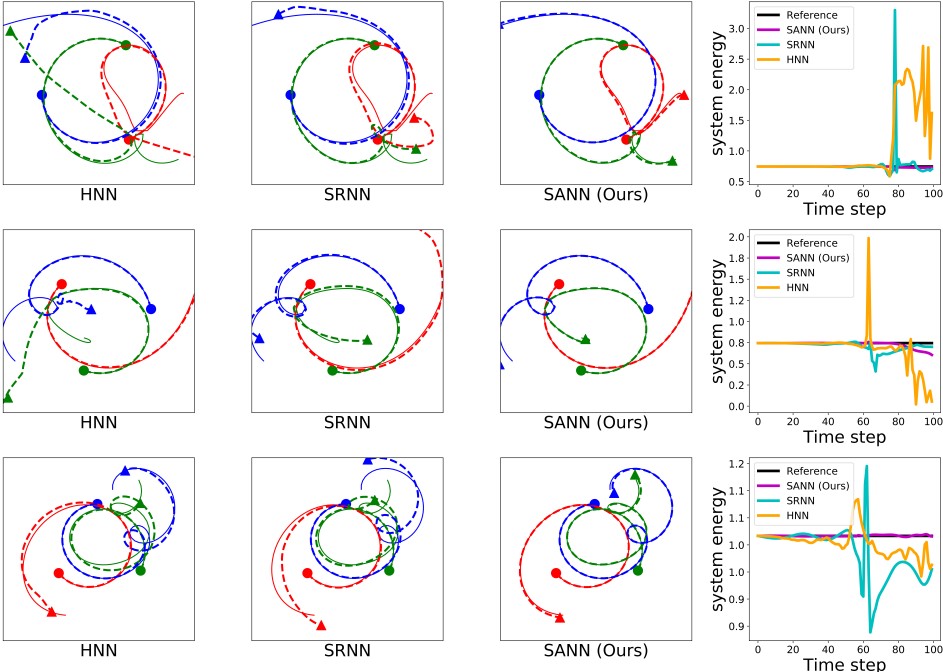

Figure 6: The first three columns show the comparison of the reference orbits (solid curves) and the orbits learned by different method (dotted curves). ● is the initial position, and ▶ is the direction. The orbit of the different particles is presented by different colors. The rightmost figures show the energy comparison. We provide more results on Appendix E and dynamic graphs in website.

## 5    STIFFNESS METRIC

In Section 3.1 SAI was introduced in order to characterize the stiffness of a trajectory. In this section we show theoretically and numerically that SAI can serve as a proxy of SI in a data-driven scenario.

**SAI keeps the same trend as SI.** With the analytic expression of dynamics in three-body problem and a given trajectory, we compute the SI and SAI for each time step of the trajectory, respectively. As a representative example, Fig. 7 shows the orbits of three particles and the corresponding curves of SAI and SI with time. We can see that SI remains roughly a small constant when three particles are far away from each other but rises sharply when two particles get close. In three-body dynam-

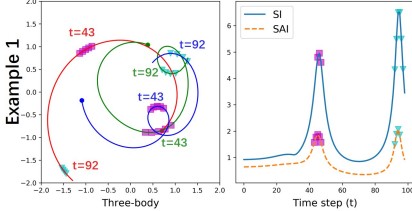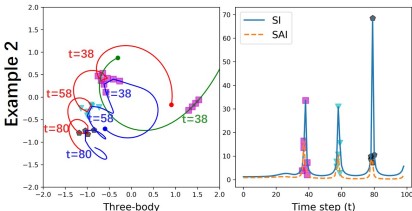

Figure 7: Comparison of SAI and SI with time. The color markers refer to the occurrence of the stiffness phenomena; the same color corresponds to the same time period. The trends of SAI and SI are consistent. We provide more results on Appendix E and dynamic graphs in website.

ics, the stiff phenomena are known to occur at the close interaction of particles, which is in good correspondence to large SI values. More important, SAI keeps the same trend with SI as time goes by, and SAI reaches the peak at the same time interval as SI does. Through extensive numerical validation like this, we find that SAI can characterize the stiffness of a trajectory very well.

**SAI is a generalized SI.** For a theoretical analysis, we follow the standard technique that linearizes the ODE locally (Arrowsmith & Place, 1992). In our method, SAI is computed by adjacent states with small step size, and the adjacent states satisfy the linearized ODE. By definition, SI depends on the eigenvalue $\lambda_m$ of the linear ODE with the maximum absolute real part. We show in Theorem 1 (with proof in Appendix D) that SAI depends on all the eigenvalues. In an ideal setting where $\ell_m = 1, \ell_i = 0, \forall i \neq m$, SAI is roughly equivalent to SI.

**Theorem 1.** *Let $A$ be a $n \times n$ symmetric real matrix and $\{\lambda_i\}_{i=1}^n$ be its $n$ distinct eigenvalues. Then for the linear system $\frac{d\boldsymbol{x}}{dt} = \boldsymbol{Ax}$ with nonzero initial condition $\boldsymbol{x}(0) = \boldsymbol{u}^0$, we have*

$$\frac{1}{\|\boldsymbol{u}^0\|_2}\Big\|\frac{\boldsymbol{u}^1 - \boldsymbol{u}^0}{\Delta t}\Big\|_2 = \Big(\sum_{i=1}^n \ell_i \lambda_i^2\Big)^{1/2} + \mathcal{O}(\Delta t) \tag{11}$$

*where $\boldsymbol{u}^1 = \boldsymbol{x}(\Delta t)$ and $\ell_i \geq 0, i = 1, ..., n$ with $\sum_{i=1}^n \ell_i = 1$.*

One limitation of our stiffness metric is that the value of SAI may change with the coordinate system. For example, a coordinate translation does not change the norm of the finite difference term in Eqn. (5), but may change the norm of the state. However, the stiffness classification in our method is not sensitive to coordinate translation because the classification relies only on the ranking (not their values) of the SAIs for the time steps in a trajectory. See Appendix B for a detailed analysis.

## 6 ABLATION STUDY

In this section we explore the effect of varied resampling replication $K$ introduced in Section 3.2 and activation functions used in $\mathcal{H}_\theta(\mathbf{p}, \mathbf{q})$. The experiments are conducted on the three-body problem. Table 1 shows the training MSE, testing MSE, relative energy error, and training time (in seconds) for each epoch. Fig. 5 shows the performance with and without resampling. Fig. 8 displays the performance of different activation functions used in the NN.

**Resampling.** From Exp. 2–Exp. 6, we see that with the increasing resampling replication for stiff portion, the learned Hamiltonian dynamics can achieve smaller errors, but the training time grows accordingly. For example, when the replication $K$ changes from 6 to 8, the testing MSE decreases by 47.61% but the training time increases by 22.72%.

**Efficiency.** In Exp. 7 where all training intervals are integrated with partition 50, we achieve a more accurate solution compared with Exp. 2 where some portions of intervals are integrated with partition 10. However, the training time increases significantly. On the other hand, despite the time increment caused by resampling, Exp. 4 achieves comparable performance to Exp. 7 but reduces over 40% of training time. To trade off between the accuracy and time cost, we can choose an appropriate resampling replication.

**Activation functions.** We compare the performance of ReLU, Tanh, and Rational (Boullé et al., 2020) used in the NN to approximate the Hamiltonian. Even though theoretically ReLU NN has a

good approximation property, it has worse performance in our problem where the dynamics involve a first-order derivative. Rational NN is slightly better than Tanh NN. According to Occam's Razor (Lattimore & Hutter, 2011), we intuitively prefer to formulate the physical system with a simple expression and avoid a complicated one such as Tanh NN, a highly nonlinear function. Rational NN, which has the form of a rational fraction, is more suitable for modeling the Hamiltonian.

| Exp. ID | Stiff Ratio $\gamma$ | Partition (nonstiff) | Partition (stiff) | Replication | Train MSE | Test MSE | Energy Error | Training Time |
|---------|----------------------|----------------------|-------------------|-------------|-----------|----------|--------------|---------------|
| Exp. 1  | 0.1 | 10 | 10 | 9 | 2.0e-6 | 0.0513 | 0.0611 | 13.52 |
| Exp. 2  | 0.1 | 10 | 50 | 1 | 2.0e-7 | 0.2173 | 0.7884 | 10.08 |
| Exp. 3  | 0.1 | 10 | 50 | 4 | 1.5e-7 | 0.0460 | 0.1026 | 19.82 |
| Exp. 4  | 0.1 | 10 | 50 | 6 | 7.4e-8 | 0.0294 | 0.0581 | 26.36 |
| Exp. 5  | 0.1 | 10 | 50 | 8 | 5.5e-8 | 0.0154 | 0.0228 | 32.35 |
| Exp. 6  | 0.1 | 10 | 50 | 9 | 4.3e-8 | 0.0234 | 0.0457 | 35.35 |
| Exp. 7  | 0.5 | 50 | 50 | 1 | 6.2e-7 | 0.0554 | 0.0900 | 34.75 |

Table 1: Performance comparison under different parameter settings. "Exp." is short for experiment.

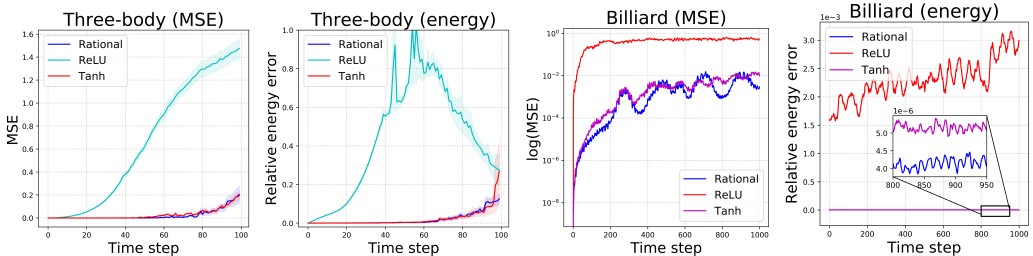

Figure 8: Performance comparison of different activation functions on the three-body problem and billiard model.

# 7    RELATED WORK

**Learning Hamiltonian system from data.** Enforcing the conservative law into the system structure becomes a powerful and popular tool to learn the Hamiltonian system from data (Willard et al., 2020; Cherifi, 2020; Zhong et al., 2021). Greydanus et al. (2019); Choudhary et al. (2020) use an NN to approximate the $\mathcal{H}(\mathbf{p}, \mathbf{q})$ instead of learning the dynamics directly. This idea also applies to learning the conserved quantities from images (Toth et al., 2020). To improve the accuracy of integration, Chen et al. (2019) conduct multi-step integration with a symplectic solver. Finzi et al. (2020) simplify the learning process by coordinate transformation and enforcing the constraints of new coordinates. To learn the Hamiltonian dynamics directly, Tong et al. (2021); Jin et al. (2020) design NNs with a symplectic structure to characterize the system, while Chen & Tao (2021) learn a symplectic map of the Hamiltonian dynamics. Zhong et al. (2020) incorporate the physical prior into the dynamics parameterization.

**Learning stiff dynamics.** It has been shown that stiffness may lead to failures in data-driven modelling (Wang et al., 2020; Kim et al., 2021; Parmar et al., 2021). Kim et al. (2021) propose a scaling strategy that mitigates the stiffness of the dynamics to stabilize the gradient calculation. This approach, however, is applicable only to problems where stiffness is caused by widely separated time scales and endures during the whole trajectory.

# 8    CONCLUSION

We propose SANN (stiffness-aware neural network) to learn the stiff Hamiltonian system. We also propose a new metric SAI (stiffness-aware index) to classify the training data into stiff and nonstiff portions. This classification along with a resampling technique allows us to apply step size adaptation strategies to better capture the dynamical characteristics of the Hamiltonian vector fields. On complex physical systems including the three-body problem and the billiard model, our method outperforms the state-of-art approaches with a significant margin. Our method has potential to extend to other types of stiff dynamical systems, not limited to learning the Hamiltonian; such an extension is left as future work.

ACKNOWLEDGEMENT

This material is based upon work supported by the U.S. Department of Energy, Office of Science, Office of Advanced Scientific Computing Research, Scientific Discovery through Advanced Computing (SciDAC) program through the FASTMath Institute under contract DE-AC02-06CH11357 at Argonne National Laboratory. S. L. acknowledges the support of the Ross-Lynn fellowship from Purdue University.

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

## A   MITIGATING IMPLICIT BIAS

In this section, we empirically demonstrate that our SANN can mitigate the implicit bias of NN-based optimization. First, the potential functions learned by different methods are compared, and we can see that SANN can learn the potential function more accurately than other methods. Then, we show the performance for excessive number of epochs to demonstrate that the baseline methods cannot capture the dynamics well even when a large number of training epochs is used.

### A.1   LEARNED POTENTIAL FUNCTION

The stiffness of the billiard model is mainly caused by the sharp boundary of the potential function. Hence, the learned potential function can reflect the accuracy of the simulated dynamics. Fig. 9 shows the top view and side view of the potential function of the billiard model from different methods. The reference potential function becomes sharp near the boundaries of the ring. From the top view, compared with SRNN and HNN, SANN learns a potential function that is close to the reference. From the side view, the potential functions of SRNN and HNN tend to learn a smooth inner circle boundary, which indicates these methods suffer from the implicit bias mentioned in Section 1. In contrast, the edge and corner of the SANN potential function appear much sharper.

### A.2   INFLUENCE OF NUMBER OF EPOCHS

Ronen et al. (2019) points out that it requires $O(k^2)$ time to learn a function of frequency $k$ for NN-based optimization. However, we observe that the extending the training time cannot mitigate the implicit bias for SRNN or HNN when learning the stiff Hamiltonian systems.

We increase the number of epochs for training SRNN and HNN from $3,000$ to $6,000$, $10,000$, and $15,000$. Fig. 10 shows the training error versus different numbers of epochs, as well as MSE and the energy error of the simulated trajectories. We can see that the training loss of SRNN converges to the same value even when the number of epochs becomes extremely large, and the MSE and the energy error do not improve much as the number of epochs grows. Similar behaviours are observed for HNN.

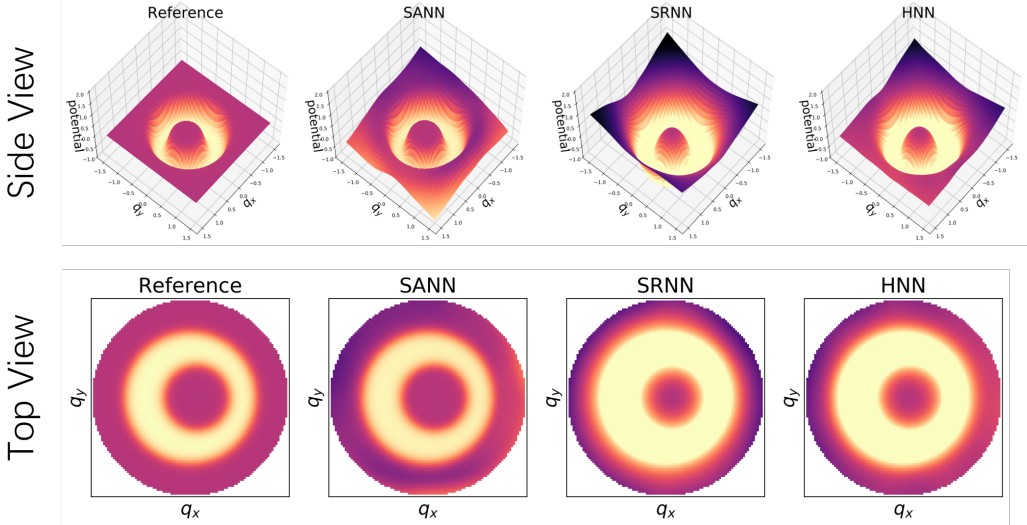

Figure 9: The learned potential function of the billiard model.

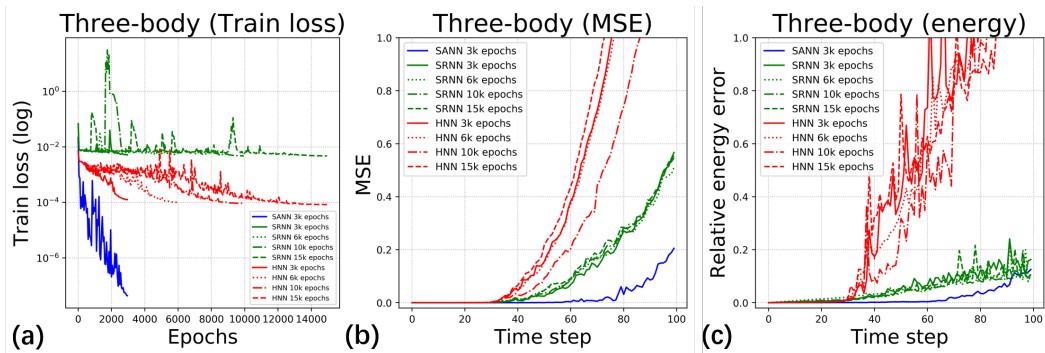

Figure 10: Comparison of different methods in training loss, MSE, relative energy error for different numbers of training epochs.

# B  GENERALITY OF SAI-BASED STIFFNESS CLASSIFICATION

In this section, we empirically demonstrate the SAI-based stiffness classification is not sensitive to the translation of the coordinate. The purpose is to show that this classification strategy can identify the stiff part of the trajectory under different kinds of coordinate translation.

Specifically, we translate the position coordinates of the three-body trajectories with two kinds of translations, fixed direction and random direction. For fixed direction, the position coordinates are added with a fixed vector $\mathbf{v} = (0, \cdots, 0, v, \cdots, v)$, e.g., $(\mathbf{p}^{t_i}, \mathbf{q}^{t_i}) + \mathbf{v}$. For a random direction, the position coordinates are added to with a random vector $\mathbf{v} = (0, \cdots, 0, v_1, \cdots, v_n)$ and $\{v_i\}$ are i.i.d random variables uniformly distributed on $[0, 1]$.

Fig. 11(a) and 11(b) highlight the stiff intervals (marked in yellow) that our method identifies using the SAIs calculated in different coordinate systems. Three examples (one in each column) are chosen for illustration. Fig. 11(a) shows the classification results ($\gamma = 0.1$) for the fixed translation using 7 widely different values of $v$ ($v = 0.1, 10, \cdots, 10^6$). Fig. 11(b) shows the classification results for 7 occurrences of the random vector $\mathbf{v}$ uniformly sampled from $[0, v]$ ($v = 0.1, 10, \cdots, 10^6$). Fig. 12 illustrates the same analysis for three additional examples.

We can see that (1) the SAI-based classification captures the stiff parts of the trajectory (those with large SI values) successfully for all scenarios, and (2) coordinate translation has almost no impact on the classification results. This is expected by construction. The second term of Eqn. (5) $\left\| \frac{\boldsymbol{u}^{t_{i+1}} - \boldsymbol{u}^{t_i}}{t_{i+1} - t_i} \right\|_2$ is a translation-invariant quantity. Coordinate translation changes only the norm of the states, so it causes the SAIs to be scaled for all the time intervals. Therefore, when sorting the intervals based on SAI, we can maintain a similar order for these intervals and identify the stiff parts accurately.

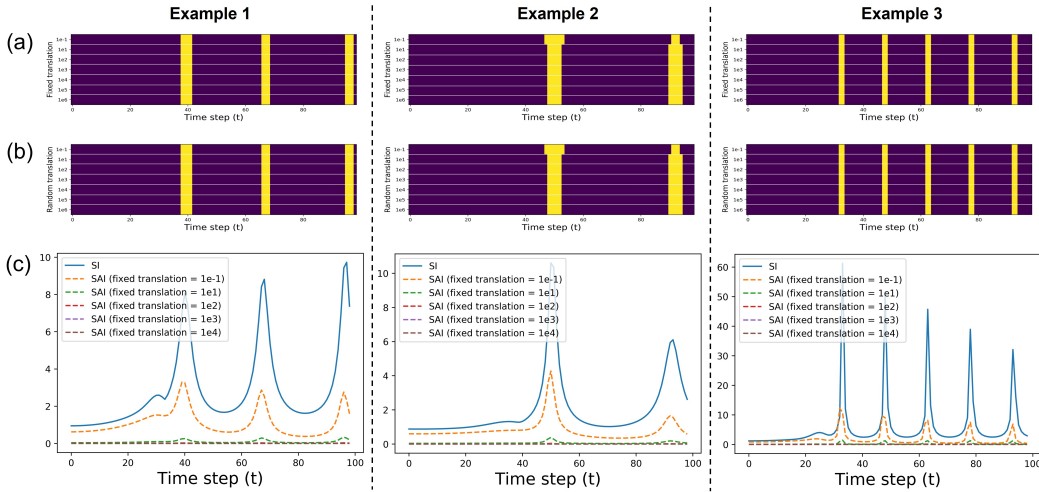

Figure 11: (a) SAI-based classification for a trajectory translated by a fixed vector for 7 different values of $v$ ($v = 0.1, 10, \cdots, 10^6$). Stiff intervals are marked in yellow. (b) SAI-based classification for a trajectory translated by a uniform random vector $\mathbf{v}$ on $[0, v]$ ($v = 0.1, 10, \cdots, 10^6$). (c) Comparison of SI, SAI of the original trajectory, and SAI of the translated trajectory.

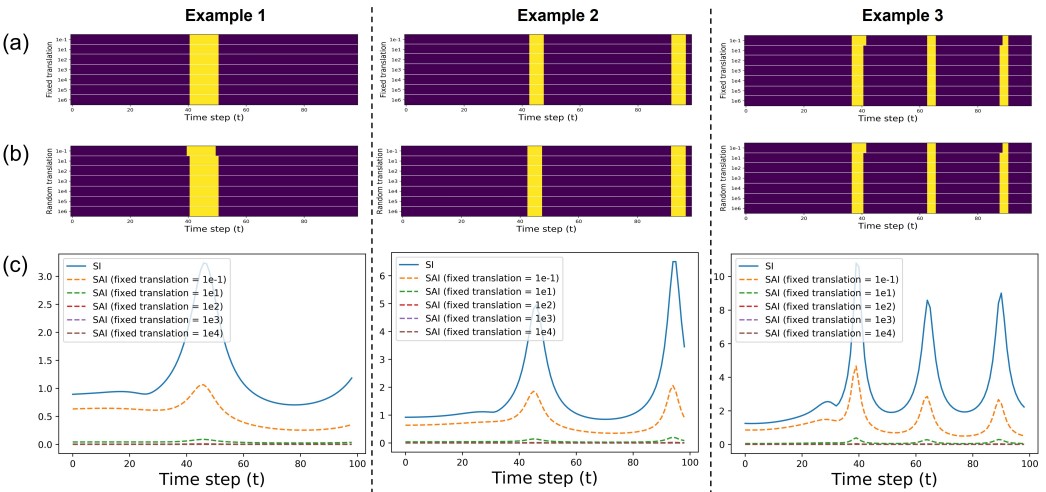

Figure 12: (a) SAI-based classification for a trajectory translated by a fixed vector for 7 different values of $v$ ($v = 0.1, 10, \cdots, 10^6$). Stiff intervals are marked in yellow. (b) SAI-based classification for a trajectory translated by a uniform random vector $\mathbf{v}$ on $[0, v]$ ($v = 0.1, 10, \cdots, 10^6$). (c) Comparison of SI, SAI of the original trajectory, and SAI of the translated trajectory.

## C  LEAPFROG INTEGRATION METHOD

The forward step of Leapfrog on separable Hamiltonian $\mathcal{H}(\mathbf{p}, \mathbf{q}) = T(\mathbf{p}) + V(\mathbf{q})$ is given as follows (Chen et al., 2019),

$$\mathbf{p}^{t_{i+\frac{1}{2}}} = \mathbf{p}^{t_i} - \frac{1}{2}(t_{i+1} - t_i)\nabla_{\mathbf{q}}V(\mathbf{q}^{t_i}),$$

$$\mathbf{q}^{t_{i+1}} = \mathbf{q}^{t_i} + (t_{i+1} - t_i)\nabla_{\mathbf{p}}T(\mathbf{p}^{t_{i+\frac{1}{2}}}),$$

$$\mathbf{p}^{t_{i+1}} = \mathbf{p}^{t_{i+\frac{1}{2}}} - \frac{1}{2}(t_{i+1} - t_i)\nabla_{\mathbf{p}}V(\mathbf{q}^{t_{i+1}}).$$

## D  PROOF OF THEOREM 1

**Theorem 1**. Let $A$ be an $n \times n$ symmetric real matrix and $\{\lambda_i\}_{i=1}^n$ be its $n$ distinct eigenvalues. Then for the linear system $\frac{d\boldsymbol{x}}{dt} = \boldsymbol{A}\boldsymbol{x}$ with nonzero initial condition $\boldsymbol{x}(0) = \boldsymbol{u}^0$, we have

$$\frac{1}{\|\boldsymbol{u}^0\|_2}\Big\|\frac{\boldsymbol{u}^1 - \boldsymbol{u}^0}{\Delta t}\Big\|_2 = \Big(\sum_{i=1}^n \ell_i \lambda_i^2\Big)^{1/2} + \mathcal{O}(\Delta t), \tag{12}$$

where $\boldsymbol{u}^1 = \boldsymbol{x}(\Delta t)$ and $\ell_i \geq 0, i = 1, ..., n$ with $\sum_{i=1}^n \ell_i = 1$.

*Proof.* Let $\{\boldsymbol{v}_i\}_{i=1}^n$ be the eigenvectors corresponding to the eigenvalues $\{\lambda_i\}_{i=1}^n$. Since $A$ is a symmetric real matrix, $\{\boldsymbol{v}_i\}_{i=1}^n$ form a set of orthogonal vectors. Without loss of generalization, we assume $\{\boldsymbol{v}_i\}_{i=1}^n$ is standard orthogonal basis. Let $\boldsymbol{x}(0) = \boldsymbol{u}^0 = \sum_{i=1}^n c_i \boldsymbol{v}_i$; then $\|(c_1, ..., c_n)\|_2 = \|\boldsymbol{u}^0\|_2$. The solution of the linear system is $\boldsymbol{x}(t) = \sum_{i=1}^n c_i \boldsymbol{v}_i e^{\lambda_i t}$. Therefore,

$$\lim_{\Delta t \to 0}\frac{1}{\|\boldsymbol{u}^0\|_2}\Big\|\frac{\boldsymbol{u}^1 - \boldsymbol{u}^0}{\Delta t}\Big\|_2 = \lim_{\Delta t \to 0}\frac{1}{\|\boldsymbol{u}^0\|_2}\Big\|\frac{\sum_{i=1}^n c_i \boldsymbol{v}_i e^{\lambda_i \Delta t} - \sum_{i=1}^n c_i \boldsymbol{v}_i}{\Delta t}\Big\|_2$$

$$= \frac{1}{\|\boldsymbol{u}^0\|_2}\Big\|\sum_{i=1}^n c_i \lambda_i \boldsymbol{v}_i\Big\|_2 \tag{13}$$

$$= \Big(\sum_{i=1}^n \frac{c_i^2}{\sum_{s=1}^n c_s^2}\lambda_i^2\Big)^{1/2}.$$

We let $\ell_i \triangleq \frac{c_i^2}{\sum_{s=1}^n c_s^2} \geq 0$ and have $\sum_{i=1}^n \ell_i = 1$. $\qquad\square$

# E    SUPPLEMENTARY RESULTS

In this section we provide some supplementary results to the experiments of the main text.

Fig. 13 gives a comparison of the predicted trajectories of the billiard model, and this is a supplement to Fig. 4. Again, we see that SANN recovers visually the same trajectories as the reference. For the three-body problem, Fig. 14 and Fig. 15 show the orbits of three particles learned by different methods, and these results are a supplement to Fig. 6. We can see that HNN and SRNN diverge from the orbits after a period of time while SANN produces orbits that are nearly identical to the orbits of the reference and the energy is roughly conserved.

Fig. 16 displays the comparison of SAI and SI with time, and these results are a supplement to Fig. 7. These four examples in Fig. 16 support that the SAI keeps the same trend as SI and achieves the peak at the same time as SI does.

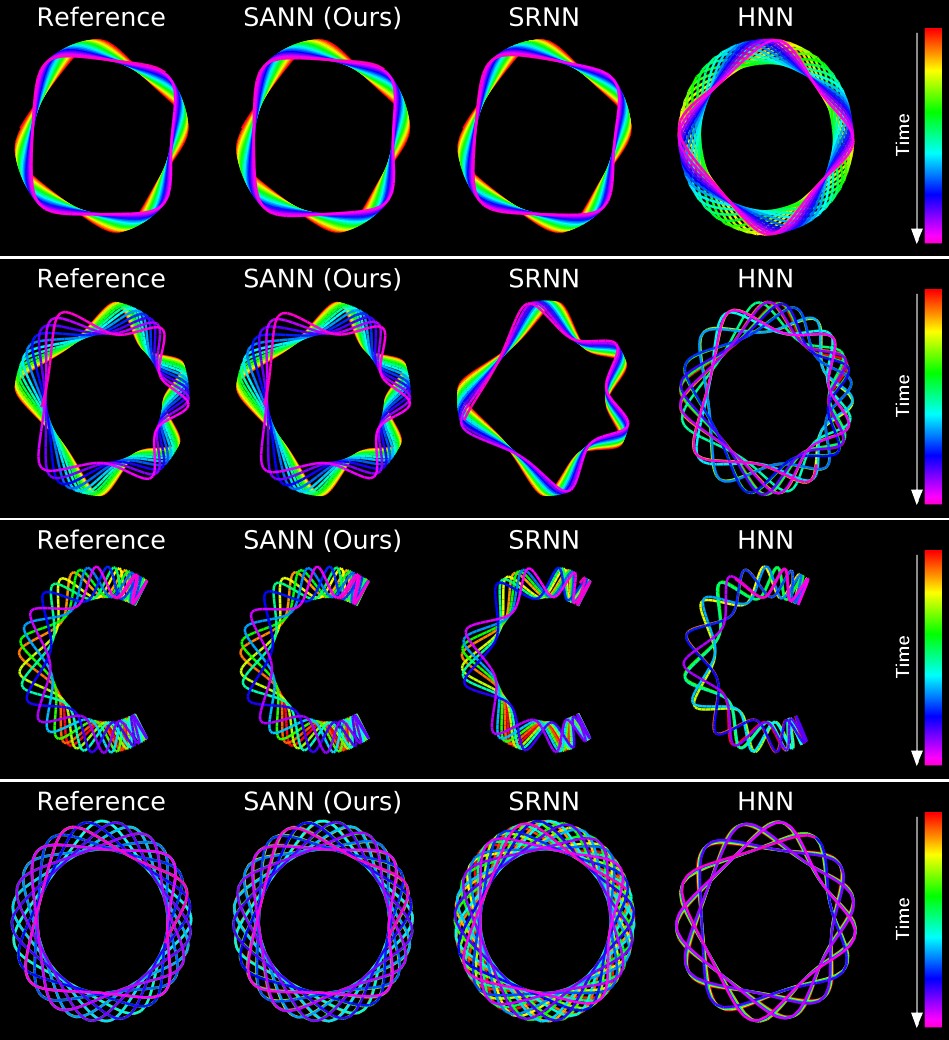

Figure 13: (Supplement to Fig. 4) Comparison of billiard orbits simulated using the Hamiltonian learned by different methods.

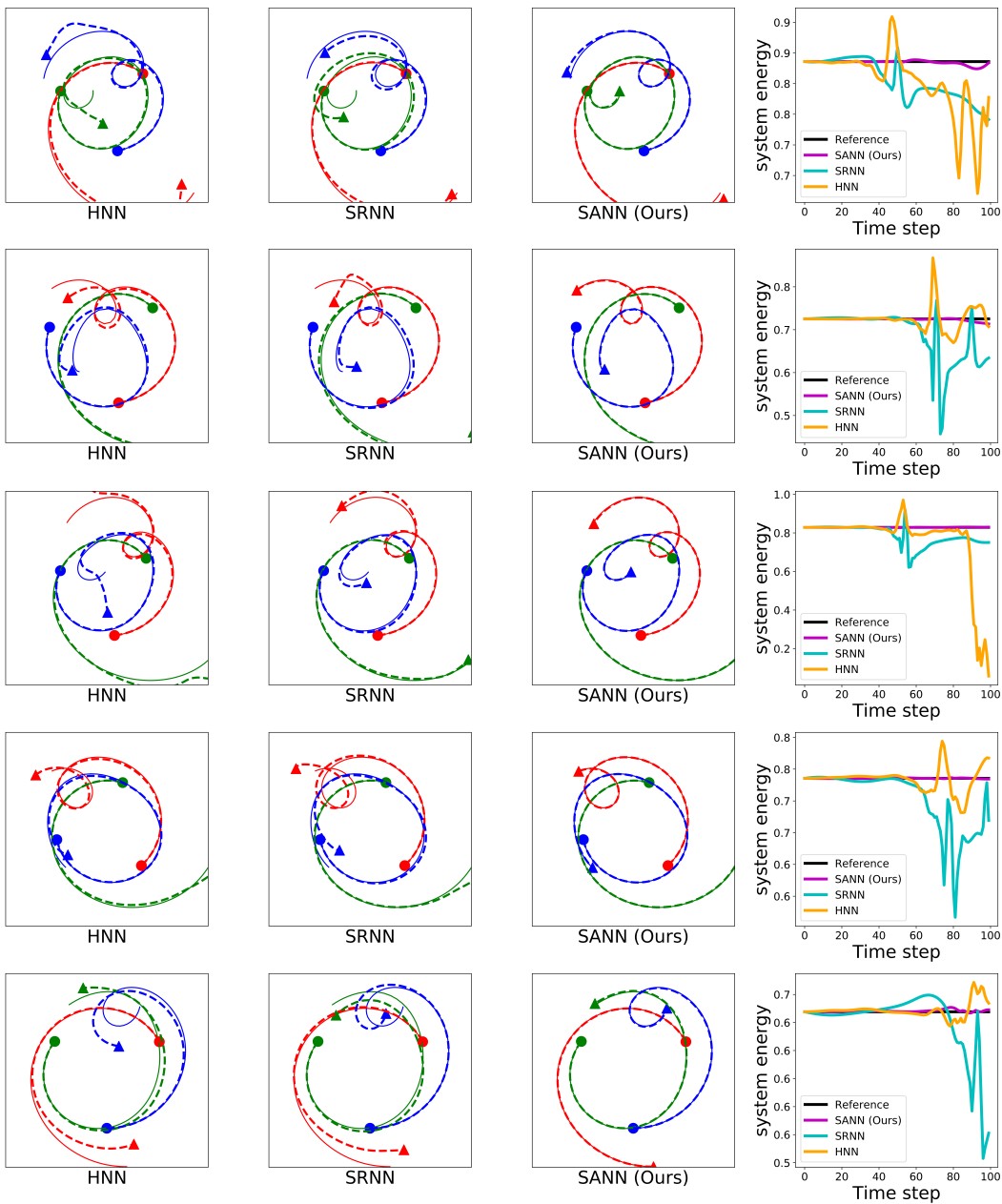

Figure 14: (Supplement to Fig. 6) The first three columns show the comparison of the reference orbits (solid curves) and the orbits learned by different method (dotted curves). ● is the initial position, and ▶ is the direction.

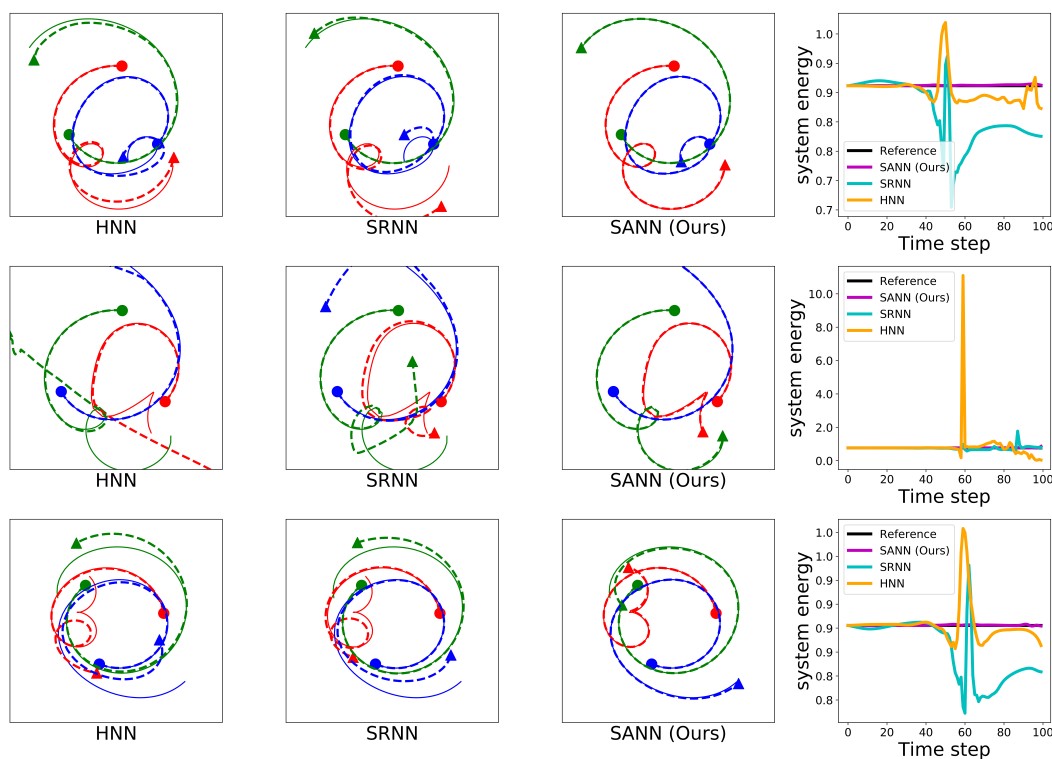

Figure 15: (Supplement to Fig. 6) The first three columns show the comparison of the reference orbits (solid curves) and the orbits learned by different methods (dotted curves). ● is the initial position, and ► is the direction.

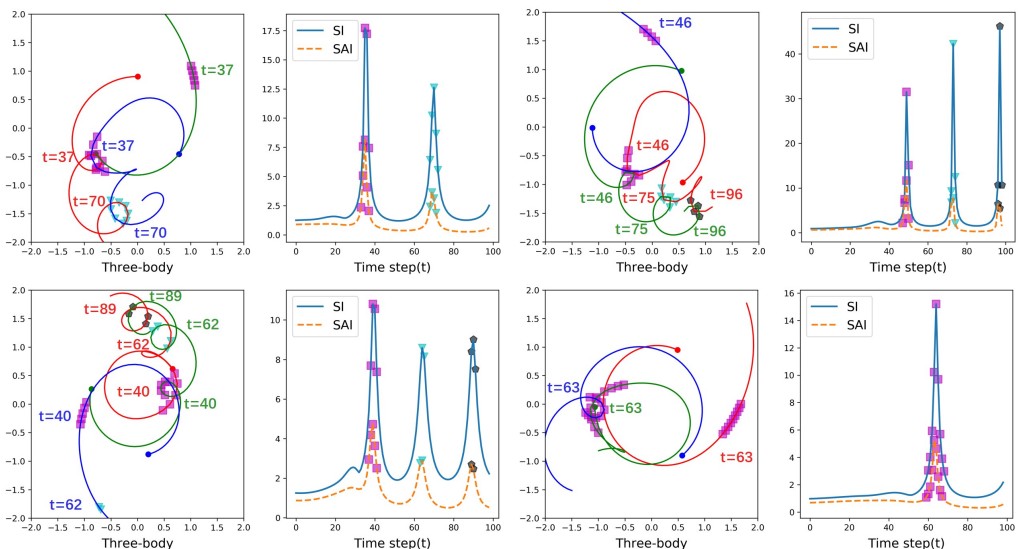

Figure 16: (Supplement to Fig. 7) Comparison of SAI and SI with time. The color markers refers to the occurrence of the stiffness phenomena, and the same color corresponds to the same time period. The trends of SAI and SI are consistent.

# F  NOISY DATA

In this section we evaluate our method on the noisy training data on Billiard model. We add the Gaussian noise with different noise levels (standard deviation) to the clean training data. Figures 17 and 18 respectively show the MSE and relative energy error with time under noisy data with different noise level. We can see that all the learned dynamical systems become unstable and the energy gets nonconserved. We see that HNN, SRNN, and SANN have difficulty in learning the chaotic dynamics when the data is disturbed by noise.

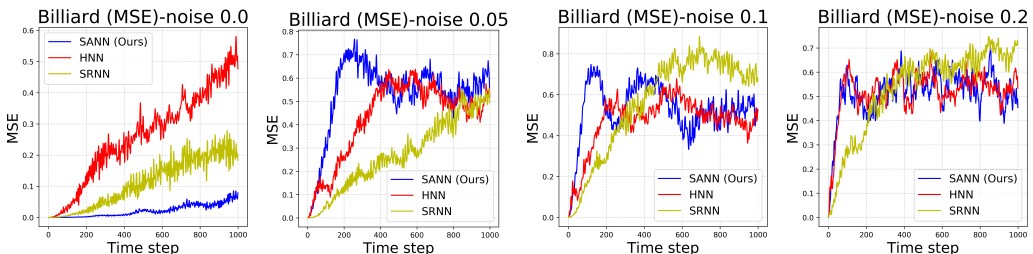

Figure 17: Comparison of MSE of the learned dynamical systems under noisy data with different noise levels.

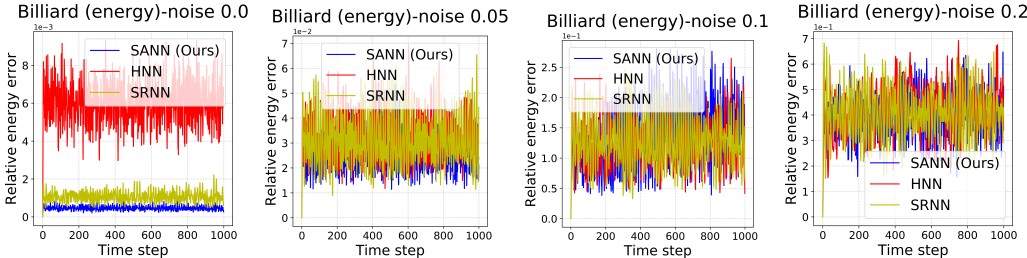

Figure 18: Comparison of relative energy error of the learned dynamical systems under noisy data with different noise levels.

## G  HYPERPARAMETER SELECTION

We determine the stiff ratio $\gamma$ based on the validation dataset. We investigate the distribution about the SAI over a trajectory and estimate the ratio of the stiff portion for the dataset. Take the dataset of the three-body problem as an example. We normalize the SAIs of a trajectory to $[0, 1]$ and show the increasing order of SAIs in Fig. 19. One can see that large SAIs mainly concentrate around the order index from 90 to 100. Hence, we choose 90-percentile ($\gamma = 0.1$) as the threshold.

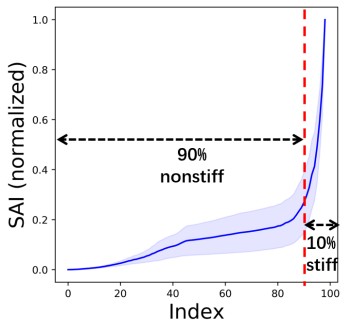

Figure 19: The increasing order of the normalized SAIs over the three-body dataset of 1,000 trajectories. The shaded area represents the 95% confidence interval.

For the partition $S$ for nonstiff and stiff portions, we perform the grid search for the pairs of parameters as did in  Choudhary et al. (2020).  We use the validation loss as a proxy of the final performance. In particular, we first conduct 10-epoch training and sort the pairs of parameters based on the validation loss. Then, we choose the parameters with the smallest validation loss.

# H    INSEPARABLE HAMILTONIAN

SANN can also be applied to inseparable Hamiltonian. Like HNN, we use an NN that takes $\mathbf{p}$ and $\mathbf{q}$ as an input to approximate the Hamiltonian directly on two problems: a chain of $N$ pendulums(Finzi et al., 2020) and the three-body problem.

In the Pendulum-N problem, let $m_i$ be the mass of the $i$-th pendulum, $\ell_i$ be the rigid rod between $(i-1)$-th and $i$-th pendulum, $p_i$ be the generalized momentum and $q_i$ be the angle between the $i$-th pendulum and the $y$-axis. The Hamiltonian can be defined as

$$\mathcal{H}(\mathbf{p}, \mathbf{q}) = \frac{1}{2}\mathbf{p}^\top \boldsymbol{M}(\mathbf{q})^{-1}\mathbf{p} - \sum_{i=1}^{N}\sum_{k=1}^{i} m_i \ell_k \cos q_k, \tag{14}$$

where the mass matrix has a complicated form $\boldsymbol{M}(\mathbf{q})_{i,j} = \cos(q_i - q_j)\ell_i\ell_j \sum_{k=\max\{i,j\}}^{N} m_k$. We consider $N = 4$ and $N = 6$, and set $g = m_i = \ell_i = 1$, $i = 1, 2, \cdots, N$. We simulate the training set with 200 trajectories and the testing set with 100 trajectories by RKF45. The initial states are uniformly and randomly sampled from $[-0.25, 0.25]$. Each trajectory contains 100 time steps with a step size of 0.03.

The three-body problem is described in Section 4.2. It has a separable Hamiltonian, but we treat it as an inseparable Hamiltonian in this experiment.

We compare SANN with HNN using these two problems. For both methods, the Euler method is used for training while RKF45 is used for predicting. For SANN, we set the partition parameter $S$ to be 50 and 10 for the $N$-chain pendulum and the three-body problem, respectively. Table 2 shows the train MSE and test MSE for SANN and HNN. We can see that SANN outperforms HNN in all the tests.

| Dataset | Method | Train MSE | Test MSE |
|---------|--------|-----------|----------|
| Pendulum-4 | HNN | 3.80E-08 | 8.40E-05 |
|            | SANN | 1.80E-08 | 4.60E-05 |
| Pendulum-6 | HNN | 1.60E-07 | 4.80E-04 |
|            | SANN | 1.00E-07 | 3.90E-04 |
| Three-body | HNN | 4.20E-06 | 1.60E+00 |
|            | SANN | 8.60E-07 | 7.00E-01 |

Table 2: Train MSE and test MSE for HNN and SANN.

