# OpenReview forum: "Stiffness-aware neural network for learning Hamiltonian systems"
_ICLR.cc/2022/Conference — ICLR 2022 Poster_

### Official Review · Reviewer_MYXe · 2021-10-24

**Correctness:** 3
**Technical Novelty And Significance:** 2
**Empirical Novelty And Significance:** 3
**Recommendation:** 6
**Confidence:** 4

**Main Review:**

### Positives

It is an insightful suggestion that the stiffness is a bottleneck to learning of a physical system.

It is surprising that a simple oversampling is enough.

### Negatives

Section 5 demonstrates that SAI is a good approximation to stiffness index (SI), but this might not hold for different coordinate systems. SI assumes that the origin of the coordinate system is an equilibrium point. In other words, the bias from the equilibrium point is already subtracted from the position. However, SAI does not (or cannot in practice). The norm of the state, and thereby, SAI depend on the coordinate system. It is preferable to valid the generality of SAI.

The sampling strategy is heuristic. As shown in Table 1, the results are sensitive to the hyperparameter tuning. A guidance justified theoretically is preferable.

### Minor Comments

It might be an insightful suggestion that the neural network tends to learn a smooth function and this implicit bias is a bottleneck to learning of a stiff system. However, this suggestion is not validated by experiments, and it is unclear whether the proposal resolves this problem. An additional experiment or analysis is not mandatory, but may improve the contribution of the present study.

Figure 1 is a bit confusing. I prefer that the scales of the axes are fixed within each example.

### After discussion

All my concerns were addressed by the additional experiments and explanations. I update the score from 5 to 6.

**Summary Of The Paper:**

This study proposes the stiff-aware index (SAI) for an ordinary differential equation and the training strategy that uses samples with large SAIs more frequently. A neural network has the implicit bias to tend to learn a smooth function. Only a limited portion of data obtained from a stiff system exhibits a rapid change, in other words, the training data is imbalanced between a gradual change and a rapid change. Hence, without SAI, it is difficult for a neural network to learn a stiff dynamics. The contribution of SAI is confirmed using a three-body problem.


**Summary Of The Review:**

This study is based on an insightful suggestion, and the proposed method is simple but effective. However, the strategy is heuristic, and the generality is unclear.

---

> ### Author Response · Authors · 2021-11-20
> **We thank the reviewer for their constructive comments! Please see our reply below.**
>
> We thank the reviewer for recognizing the insights of this work. In the below, we respond to the questions point by point.
>
> >Section 5 demonstrates that SAI is a good approximation to stiffness index (SI), but this might not hold for different coordinate systems. SI assumes that the origin of the coordinate system is an equilibrium point. In other words, the bias from the equilibrium point is already subtracted from the position. However, SAI does not (or cannot in practice). The norm of the state, and thereby, SAI depends on the coordinate system. It is preferable to valid the generality of SAI.
>
> We have added more results in *Appendix B* to empirically demonstrate that our method based on SAI is not sensitive to the translation of the coordinate. Note that we use SAI as a proxy for SI, but not necessarily an accurate approximation to SI. SANN can identify the stiff part of the trajectory if SAI captures the trend of SI (see the new Figures 11 and 12).
>
> >The sampling strategy is heuristic. As shown in Table 1, the results are sensitive to the hyperparameter tuning. A guidance justified theoretically is preferable.
>
> We agree with the reviewer. It is common to see that the performance depends on the hyperparameters. We followed the standard approach to find suitable values for the hyperparameters using the validation dataset. We have added *Appendix G* to provide a guidance for hyperparameter selection.
>
> >It might be an insightful suggestion that the neural network tends to learn a smooth function and this implicit bias is a bottleneck to learning of a stiff system. However, this suggestion is not validated by experiments.
>
> Thank you for this insightful suggestion. We have provided additional analysis in *Appendix A* to demonstrate this suggestion. You are right that SRNN and HNN tend to learn a smooth potential function as shown in Figure 9.
>
> >Figure 1 is a bit confusing. I prefer that the scales of the axes are fixed within each example.
>
> We have fixed the axes of Figure 1.

---

> > ### Comment · Reviewer_MYXe · 2021-11-20
> > **Response**
> >
> > Thank you for the detailed response.
> >
> > *General*:
> > Some Appendices are not cited from the body text.
> >
> > *Appendix B*:
> > The added experiments show that a small coordinate transformation has a small (but definite) impact on the SAI. A large coordinate transformation is naturally expected to have a large impact. How large a coordinate transformation would destroy the SAI? Is such a large coordinate transformation practically impossible? I believe that a sincere discussion about its limitations will rather enhance the value of the proposed method. I also think that the discussion should be in the body text even if the results are provided in Appendix.
> >
> > Appendix A is very impressive and suggestive of further applications.

---

> > > ### Author Response · Authors · 2021-11-22
> > > **We thank the reviewer for the quick reply! Please see our additional reply below.**
> > >
> > > >General: Some Appendices are not cited from the body text.
> > >
> > > Thanks for catching this. We have cited all the newly added Appendices in the body text.
> > >
> > > >Appendix B: The added experiments show that a small coordinate transformation has a small (but definite) impact on the SAI. A large coordinate transformation is naturally expected to have a large impact. How large a coordinate transformation would destroy the SAI? Is such a large coordinate transformation practically impossible? I believe that a sincere discussion about its limitations will rather enhance the value of the proposed method. I also think that the discussion should be in the body text even if the results are provided in Appendix.
> > >
> > > We see how Appendix B could cause confusion for readers. So we have made the following changes in the updated version:
> > > - We rewrote Appendix B carefully, extended the experiments using large coordinate translation, and improved Figures 11 and 12. We clarified that the stiffness classification is not sensitive to coordinate translation because the stiff part of a trajectory is identified based on the ranking of SAIs (not their values).
> > > - We added in the body text Section 5 Stiffness Metric a discussion about the limitation of the stiffness metric as suggested, and we made it clear that this has very little impact on the classification results because the proposed method relies on the ranking of SAIs.

---

> > > > ### Comment · Reviewer_MYXe · 2021-11-24
> > > > **Response**
> > > >
> > > > Thank you for your update. I understood your proposal more accurately. I am inclined to raise the score if AC allows.

---

### Official Review · Reviewer_RqeB · 2021-11-02

**Correctness:** 3
**Technical Novelty And Significance:** 3
**Empirical Novelty And Significance:** 3
**Recommendation:** 6
**Confidence:** 3

**Main Review:**

Separating the stiff and non-stiff parts of the data when training a Hamiltonian network is a novel idea. And the results show certain advantages of using this method. And experiments are included to discuss the influence of resampling, integration partitions, and activation functions.
However, only two examples are shown in the experiments, it would be more convincing if experiments of more Hamiltonian systems can be conducted.

Some additional questions:
When predicting future states, does the model use s fixed time step? (or still using a smaller timestep for stiff parts? If so, how is the stiffness calculated?)
Are there significant advantages of using an additional trainable term p^TMp, instead of training a single network that represents p^TMp + φ(q;W)?



**Summary Of The Paper:**

This paper proposes to improve the learning of a Hamiltonian system, by characterizing the stiffness of the time series data.

A stiffness-aware index is first used to classify the time interval into stiff and nonstiff. Then, during the training of the Hamiltonian network, the stiff part is integrated using a smaller timestep, and also sampled more frequently in the training data.

**Summary Of The Review:**

There are certain contributions of this paper on proposing characterizing the stiffness of the time series data, which shows its advantage of improving the performance of the Hamiltonian network.

---

> ### Author Response · Authors · 2021-11-20
> **We thank the reviewer for their insightful comments! Please see our reply below.**
>
> We thank the reviewer for acknowledging the contributions of this paper. In the below, we respond to the questions point by point.
>
> >However, only two examples are shown in the experiments, it would be more convincing if experiments of more Hamiltonian systems can be conducted.
>
> Thanks for your suggestion. We have done additional experiments on the Pendulum-N problem, which involves a non-diagonal mass matrix and inseparable Hamiltonian. Please see the newly added *Appendix H Inseparable Hamiltonian*.
>
> >Some additional questions: When predicting future states, does the model use a fixed time step? (or still using a smaller timestep for stiff parts? If so, how is the stiffness calculated?)
>
> For a fair comparison, we use the fixed step size for all the methods when predicting the future states.
>
> > Are there significant advantages of using an additional trainable term $p^T M p$, instead of training a single network that represents $p^TMp + φ(q;W)$?
>
> First, the separability of Hamiltonian only affects the choice of time integration scheme in the proposed framework. For example, we can use symplectic integrators for separable Hamiltonian to achieve higher accuracy and better stability. One can choose other integration strategies if the Hamiltonian is not separable. This is also pointed out in the SRNN paper (Chen et al., 2019, footnote at page 2): “Extending this work to non-separable Hamiltonians can be achieved by rewriting the numerical integration schema using an extended phase space (Tao, 2016).”
>
> Second, using the additional trainable term allows us to incorporate more physical information in the learning framework. This technique has also been used in the following two papers that we have cited:
>
> 1. Zhong et al. Symplectic ODE-Net:  Learning Hamiltonian Dynamics with Control. ICLR , 2020
> 2. Zhong et al. Benchmarking energy-conserving neural networks for learning dynamics from data, PMLR, 2021

---

### Official Review · Reviewer_MCLP · 2021-11-05

**Correctness:** 3
**Technical Novelty And Significance:** 3
**Empirical Novelty And Significance:** 3
**Recommendation:** 6
**Confidence:** 3

**Main Review:**

Strong points.
- A stiffness-aware approach for learning Hamiltonian systems is novel.
- The experiments show that SANN can accurately simulate complex Hamiltonian systems than baseline methods, HNN and SRNN.
- This paper is well-written.

Weak points.
- There are some hyper-parameters to be determined manually.
- The compared methods are the bare minimum.
- There are several concerns about the experimental setting.

Comments.
1. The task of learning physical dynamics from data has been of great interest recently. A key idea of incorporating stiffness into the learning scheme is novel and exciting. The proposed method is simple but effective; however, there are the hyper-parameters $\gamma, S$ to be manually determined. Especially, the ratio of the stiff portion, $\gamma$, could be critical for performance. Could the authors explain how to estimate this hyper-parameter for various physical systems, including the $M$-body problem when $M$ is large?

2. The proposed method classifies the intervals into binary labels, that is, stiff or nonstiff. It might be helpful to model the continuous stiff-ratio for each interval that is learnable.

3. The authors assume the separable Hamiltonian $H(p,q)=T(p)+V(q)$. Can the proposed learning framework apply to the inseparable generalized model?

4. The HNN (Greydanus et al., 2019) does not consider the separable assumption. In the experiments, did the authors use this assumption in the HNN learning? Also, The input of HNN is the partial derivatives, unlike SANN and HRNN. How did the authors give the input to HNN?

5. Why did the authors use the Leapfrog solver? Isn't the simple Euler method appropriate?

**Summary Of The Paper:**

This paper presents a new method, stiffness-aware neural network (SANN), for learning Hamiltonian systems from data. The authors define a stiffness-aware index (SAI) for classifying the training data into stiff and nonstiff samples. Based on the classification result, the step size for the numerical solver is adjusted, and the number of samples for training is balanced. The effectiveness of SANN is demonstrated using chaotic Hamiltonian systems, i.e., a three-body problem and billiard model.

**Summary Of The Review:**

This paper addresses the interesting problem and is well-motivated. Also, the proposed approach is novel, and the experimental results are insightful. Although I tend to accept this paper, I have some concerns, which I detail in the main review.

---

> ### Author Response · Authors · 2021-11-20
> **We thank the reviewer for their considerate comments! Please see our reply below.**
>
> We thank the reviewer for confirming the novelty and significance of this work. In the below, we respond to the questions point by point.
>
> >There are the hyper-parameters to be manually determined. Especially, the ratio of the stiff portion, $\gamma$, S could be critical for performance. Could the authors explain how to estimate this hyper-parameter for various physical systems, including the M-body problem when M is large?
>
> It is common to see that the performance depends on the hyperparameters. We followed the standard approach to find suitable values for the hyperparameters using the validation dataset. We have added Appendix G Hyperparameter selection to provide more details.
>
> >It might be helpful to model the continuous stiff-ratio for each interval that is learnable.
>
> Thank you for bringing up this interesting idea. It would be a good direction for future work. There are two challenges associated with modeling the continuous stiff-ratio.
> 1. One needs to consider how to update the learnable ratio such that it helps improve accuracy.
> 2. The extra trainable parameters may not only increase the computational cost but also add additional complexities to the algorithm.
>
> >The authors assume the separable Hamiltonian $h(p,q)=T(p)+V(q)$. Can the proposed learning framework apply to the inseparable generalized model?
>
> Yes, the proposed method can also be applied to inseparable Hamiltonian. We have done some additional experiments for inseparable Hamiltonian and included the results in *Appendix H Inseparable Hamiltonian*.
>
> Actually the separable assumption only affects the choice of time integration scheme in the proposed framework. One can choose other integration strategies if the Hamiltonian is not separable. This is also pointed out in the SRNN paper (Chen et al., 2019, footnote at page 2): "Extending this work to non-separable Hamiltonians can be achieved by rewriting the numerical integration scheme using an extended phase space (Tao, 2016)."
>
> >In the experiments, did the authors use this assumption in the HNN learning? Also, The input of HNN is the partial derivatives, unlike SANN and SRNN. How did the authors give the input to HNN?
>
> In the HNN learning, we only use the separable assumption when predicting with a symplectic integrator, and we use the same integrator for all the learning approaches when predicting for a fair comparison. You are right that the partial derivatives of the state are not available. We used finite differences to approximate the partial derivative, as did in the HNN paper (section 3.1 in Greydanus et al., 2019).
>
> >Why did the authors use the Leapfrog solver? Isn't the simple Euler method appropriate?
>
> Symplectic integrator is a numerical method widely used for integrating the Hamiltonian systems [1,2,3,4] as it can conserve energy, and it is usually more stable than non-symplectic integrators. The Euler method is a first-order method. In contrast, the Leapfrog method is a simple second-order symplectic method with better stability. [1] has numerically demonstrated that using the Leapfrog method in both training and prediction gives better accuracy than using the Euler method (see Fig. 2 in [1]).
> 1. Chen et al, Symplectic Recurrent Neural Networks, ICLR 2020
> 2. Tong et al, Symplectic neural networks in Taylor series form for Hamiltonian systems, Journal of Computational Physics, 2021
> 3. DiPietro et al, Sparse Symplectically Integrated Neural Networks, NeurIPS 2020
> 4. Jin et al, SympNets: Intrinsic structure-preserving symplectic networks for identifying Hamiltonian systems, Neural Networks 2020

---

### Official Review · Reviewer_iTTU · 2021-11-06

**Correctness:** 3
**Technical Novelty And Significance:** 3
**Empirical Novelty And Significance:** 3
**Recommendation:** 8
**Confidence:** 4

**Main Review:**

The paper is well-written, and it conveys the key ideas in a very precise manner. The authors have done an excellent job in highlighting the need for a stiffness-aware approach. Figure 1 and the related discussions in the Introduction section provide a compelling motivation for the problem.

The idea is straightforward but quite effective, as one can see from the experimental results. However, unless some aspects of the paper are further improved or better explained, it remains challenging to evaluate the real contribution and impact of this work.

The paper mentions that the mass matrix ($M$) is diagonal and subsequently it uses a set of trainable parameters to learn the individual elements of this diagonal matrix. Would you please confirm if this is indeed the case or the writing is conveying a wrong picture? For a large class of Hamiltonian systems, the mass/inertia matrix is not diagonal, and its entries depend on the generalized coordinates; for example, please consider a $k$-link pendulum with joint angles as the generalized coordinates. Therefore, if $M$ is indeed assumed to be a diagonal matrix, the scope of this work is very restricted unless the authors can show with additional experiments that the proposed approach holds true even when the mass matrix is non-diagonal and position-dependent.

Basri et al. (*The Convergence Rate of Neural Networks for Learned Functions of Different Frequencies*, NeurIPS 2019) have demonstrated that the number of epochs to learn a function has a squared relationship with its frequency. As the stiffness of an ODE is related to the presence of very high-frequency components in its solution, it may be possible that SRNN or HNN can achieve comparable accuracy if they are trained over a sufficiently long period. But the current set of experiments do not provide a precise answer to this equation. Therefore, I would strongly encourage the authors to run an additional experiment that trains SRNN and HNN with a very high number of epochs.

Finzi et al. (*Simplifying Hamiltonian and Lagrangian Neural Networks via Explicit Constraints*, NeurIPS 2020) have shown that the $L_1$ loss functions exhibit better performance while inferring dynamics from data. It would be helpful to see if the same holds true for this problem as well. Therefore, the authors should consider carrying out an additional ablation study that shows how the performance varies when the loss function uses the $L_1$ norm.

Recent work by Kim et al. (*Stiff Neural Ordinary Differential Equations*, arXiv:2103.15341) has extended the scope of Neural ODEs to stiff differential equations. I would encourage the authors to consider this approach (if possible) as an additional baseline.

The Related Work section has missed some relevant prior work that enforces Hamiltonian dynamics while using a neural network to infer dynamics from data. Please refer to the following survey papers and the references therein for further details about the relevant prior work: *Integrating Physics-Based Modeling with Machine Learning: A Survey* (Willard et al., arXiv:2003.04919), *An overview on recent machine learning techniques for Port Hamiltonian systems* (Cherifi, Physica D, 2020), and *Benchmarking Energy-Conserving Neural Networks for Learning Dynamics from Data* (Zhong et al., L4DC 2021).

Some additional minor comments:

--- The second paragraph of the Introduction mentions: "the particles deviate from the reference orbits rapidly after collision". As two particles cannot collide in this setting, the authors should rephrase the sentence with "after a close encounter".

--- The authors should consider introducing $gamma$, i.e., the hyperparameter that denotes the stiffness ratio threshold, as a percentile number. It would make the point clearer.

--- Also, "the larger(stiffer)" should be either "a larger(stiffer)" or "the largest(stiffest)".

**Summary Of The Paper:**

Incorporating Hamiltonian dynamics based inductive bias into deep neural networks has gained significant attention in the machine learning community over the last few years. However, in many real physical systems, the underlying ODEs can exhibit stiffness, i.e., numerical instability, over some time intervals or for certain values of initial conditions and/or parameter choice. As a result, the existing realizations of Hamiltonian-preserving neural networks underperform in these scenarios. This paper proposes a solution to overcome this issue. In particular, it introduces an easy to compute stiffness-aware index to split the training data points into stiff and nonstiff groups, which are then treated with different integration schemes (with different values for integration time interval and time steps). When demonstrated on relevant physical systems, the proposed approach shows improved accuracy in prediction and energy conservation.

**Summary Of The Review:**

This paper has proposed a straightforward but effective solution that can infer Hamiltonian dynamics even when the governing ODEs exhibit numerical stiffness. The problem is very well-motivated, and the paper explains the core ideas in a very precise way. However, I have mentioned in the *main review*, some aspects (especially the assumption that $M$ is a diagonal matrix with entries that are position-independent) should be properly addressed or explained to increase the concreteness and overall clarity of the paper.

********** Post Rebuttal Response **********
I would like to thank the authors for addressing the prior concerns. I have updated my score.

---

> ### Author Response · Authors · 2021-11-20
> **We thank the reviewer for their thoughtful comments! Please see our reply below.**
>
> We thank the reviewer for liking our motivation and the core ideas. In the below, we respond to the questions point by point.
>
> >The paper mentions that the mass matrix (M) is diagonal and subsequently it uses a set of trainable parameters to learn the individual elements of this diagonal matrix ….
> The scope of this work is very restricted unless the authors can show with additional experiments that the proposed approach holds true even when the mass matrix is non-diagonal and position-dependent.
>
> Thanks for pointing out this problem. It is our oversight in writing. We have fixed it now. In our method, we do not make any assumption on the structure of the mass matrix. We treat $\mathbf{M}$ as a dense trainable matrix. In our implementation of the three-body problem and the billiard model, we learn a dense matrix $\mathbf{M}$ in Eqn.2 instead of a diagonal matrix $\mathbf{M}$.
>
> With this being said, we have also added new experiments on a chain of N pendulums, which has a non-diagonal mass matrix. Please see *Appendix H Inseparable Hamiltonian*. The results show that the proposed approach holds true even when the mass matrix is non-diagonal and position-dependent.
>
> >Basri et al. have demonstrated that the number of epochs to learn a function has a squared relationship with its frequency …  I would strongly encourage the authors to run an additional experiment that trains SRNN and HNN with a very high number of epochs.
>
> We have added additional experiments where we train SRNN and HNN with a large number of epochs. Please find the results in *Appendix A2 Influence of number of epochs*. The number of epochs for training SRNN and HNN is extended from 3,000 to 6,000, 10,000, and 15,000. The results (Figure 10) show that increasing the number of epochs does not improve the performance of SRNN or HNN. Thanks for suggesting these additional experiments. They indeed further confirm the advantage of SANN.
>
> >Recent work by Kim et al. (Stiff Neural Ordinary Differential Equations, arXiv:2103.15341) has extended the scope of Neural ODEs to stiff differential equations. I would encourage the authors to consider this approach (if possible) as an additional baseline.
>
> Thanks for the pointer. We have added a discussion about this paper in Section 7 Related Work to explain why the scaling technique in Kim et al. 2021 is not suitable for our problems. Note that Kim et al. 2021 address stiff problems where the stiffness is caused by widely separated time scales (which is common in chemical reaction systems) and the stiffness endures the entire trajectory. In our case, stiffness occurs for a different reason. Taking the three-body problem for example, the dynamics becomes stiff only when the particles get close.
>
> >The Related Work section has missed some relevant prior work that enforces Hamiltonian dynamics while using a neural network to infer dynamics from data.
>
> Thanks for your suggestion. We have modified this section and added the suggested references.
>
> > - As two particles cannot collide in this setting, the authors should rephrase the sentence with "after a close encounter".
> > - The authors should consider introducing gamma ...
> > - Also, "the larger(stiffer)" should be either "a larger(stiffer)" or "the largest(stiffest)".
>
> We have made the suggested corrections in the revision.

---

### Decision · Program_Chairs · 2022-01-20

**Decision:**

Accept (Poster)

**Comment:**

This paper introduces the Stiffness-aware neural network (SANN) for improving numerical stability in Hamiltonian neural networks. To this end, the authors introduce the stiffness-aware index (SAI) to classify time intervals into stiff and non-stiff portions, and propose to adapt the integration scheme accordingly.

The paper initially received three weak accept and one weak reject recommendations. The main limitations pointed out by reviewers relate to missing references from the literature, assumptions behind the proposed approach (e.g. structure of the mass matrix, separable Hamiltonian), and clarifications on experiments including additional baselines and hyper-parameter settings.
The rebuttal did a good job in answering reviewers' concerns: RiTTU increased his rating to a clear accept, and RMYXe increased his rating to weak accept.
Eventually, there is a consensus among reviewers to accept the paper.

The AC's own readings confirmed the reviewers' recommendation. The method is straightforward yet effective, and the paper is well written. The effectiveness of the proposed approach is shown in different contexts. Since several complex systems exhibit chaotic characteristics, the paper brings a meaningful contribution to the community.